# “Time Does Not Heal All Wounds”: Sexual Victimisation Is Associated with Depression, Anxiety, and PTSD in Old Age

**DOI:** 10.3390/ijerph19052803

**Published:** 2022-02-28

**Authors:** Anne Nobels, Gilbert Lemmens, Lisa Thibau, Marie Beaulieu, Christophe Vandeviver, Ines Keygnaert

**Affiliations:** 1International Centre for Reproductive Health, Department of Public Health and Primary Care, Ghent University, 9000 Ghent, Belgium; lisa.thibau@ugent.be; 2Department of Psychiatry, Ghent University Hospital, 9000 Ghent, Belgium; gilbert.lemmens@uzgent.be; 3Department of Head and Skin–Psychiatry and Medical Psychology, Ghent University, 9000 Ghent, Belgium; 4School of Social Work, Research Chair on Mistreatment of Older Adults and Research Centre on Aging, University of Sherbrooke, Sherbrooke, QC J1K 2R1, Canada; marie.beaulieu@usherbrooke.ca; 5Department of Criminology, Criminal Law and Social Law, Ghent University, 9000 Ghent, Belgium; christophe.vandeviver@ugent.be

**Keywords:** sexual violence, sexual assault, internalising disorders, trauma, older persons

## Abstract

Sexual violence (SV) has an important impact on mental health. Childhood sexual abuse is linked to internalising disorders in later life. In older adults, SV occurs more often than previously believed. Moreover, health care workers lack the skills to address SV in later life. Studies researching the mental health impact of lifetime SV, i.e., SV during childhood, adulthood, and old age, are lacking. Between July 2019 and March 2020, 513 older adults living in Belgium participated in structured face-to-face-interviews. Selection occurred via a cluster random probability sampling with a random walk finding approach. Depression, anxiety, and posttraumatic stress syndrome (PTSD) were measured using validated scales. Participants were asked about suicide attempts and self-harm during their lifetime and in the past 12 months. SV was measured using behaviourally specific questions based on a broad SV definition. We found rates for depression, anxiety, and PTSD of 27%, 26%, and 6% respectively, while 2% had attempted suicide, and 1% reported self-harm in the past 12 months. Over 44% experienced lifetime SV and 8% in the past 12 months. Lifetime SV was linked to depression (*p* = 0.001), anxiety (*p* = 0.001), and PTSD in participants with a chronic illness/disability (*p* = 0.002) or no/lower education (*p* < 0.001). We found no link between lifetime SV and suicide attempts or self-harm in the past 12 months. In conclusion, lifetime SV is linked to mental health problems in late life. Tailored mental health care for older SV victims is necessary. Therefore, capacity building of professionals and development of clinical guidelines and care procedures are important.

## 1. Introduction

Sexual violence (SV) [1] is increasingly considered an important public health problem of major societal concern [2]. In older adults, SV has been studied in the broader context of elder abuse and neglect for many years. Moreover, most studies only included hands-on SV (e.g., unwanted kissing, rape), whereas hands-off SV (sexual harassment without physical contact) was not studied. Therefore, prevalence numbers have been underestimated for a long time [3]. A recent study, which assessed SV independently from other forms of violence and used a broad definition of SV encompassing both hands-off and hands-on SV, showed that SV in older adults occurs more frequently than previously believed [4].

In older adults, mental health problems are common and lead to impaired functioning in daily life [5]. Research has shown that up to one in three older adults reports depressive symptoms [6], and up to 14% suffers from an anxiety disorder [7] and 3% from PTSD [8]. Furthermore, older adults complete suicide proportionally more often than any other age group [9]. Physical and verbal abuse against older adults have been associated with adverse mental health outcomes [10]. When it comes to SV, studies tend to focus only on the long-term mental health outcomes of child sexual abuse (CSA), indicating that CSA victims suffer more from internalising disorders in later life [11,12]. However, studies taking a life-course perspective and researching the mental health impact of lifetime sexual victimisation, including SV that happened during childhood, adulthood, and old age, are currently lacking. Several risk factors have been associated with both elder abuse and neglect and adverse mental health outcomes in late life, such as being female, increasing age, low education level, financial difficulties, suffering from a chronic illness or disability, being care dependent, low social support, and lacking resilience [7,8,9,13,14]. Both being resilient and having a high level of social support have already shown to decrease the mental health impact of sexual victimisation in younger populations [15,16,17,18]. Moreover, identifying as non-heterosexual has been associated specifically with SV in older adults and has also been linked to mental health problems in older age [4,19]. Additionally, a recent study showed that sexual victimisation remains largely undetected by mental health care workers [20]. Although the World Psychiatric Association (WPA) petitioned a routine inquiry on SV in all psychiatric assessments [21], health care workers lack the appropriate communication skills to adequately address SV in later life. They are worried that talking about SV will break down the victims’ defence mechanisms and that they will feel shocked and helpless when confronted with SV disclosure by older victims [22].

In order to provide tailored care to older victims of SV, a better understanding of the mental health impact of lifetime sexual victimisation is needed. To our knowledge, this study is the first to assess the impact of lifetime sexual victimisation on mental health outcomes in a representative sample of older adults. We used a broad definition of SV and included sexual victimisation across the life course, including SV that happened during childhood, adulthood, and old age. This study was a part of the first gender- and age-sensitive SV prevalence study in Belgium, which aimed for a better understanding of the mechanisms, nature, magnitude, and impact of SV [23]. The objectives of this paper are three-fold: (1) to establish the prevalence of depression, anxiety, PTSD, suicide attempts, and self-harm and its moderators in older adults in Belgium; (2) to research whether lifetime sexual victimisation is associated with depression, anxiety, PTSD, suicide attempts, and self-harm in older adults; (3) to test the moderating effect of the previous identified moderators of elder abuse and neglect, SV, and mental health problems (i.e., past 12-months SV, sociodemographic characteristics, health status, social support, and resilience) in the relationship between lifetime sexual victimisation and depression, anxiety, and PTSD in older adults. Based on our results, we formulate recommendations for future research and health care practices.

## 2. Materials and Methods

### 2.1. Sample Selection

Between 8 July 2019 and 12 March 2020, a representative sample of the Belgian older population participated in a structured face-to-face interview. Participants were selected using a cluster random probability sampling with a random walk finding approach [24]. They had to live in Belgium; speak Dutch, French, or English; be at least 70 years old; and have sufficient cognitive ability to complete the interview. Cognitive status was not formally tested, but it was assessed based on the ability to maintain attention during the interview and the consistency of the participant’s answers, by means of a control question comparing the participant’s birth year and age. Structured face-to-face interviews were carried out in private at the participant’s place of residence by trained interviewers. Both older adults living in the community and living in nursing homes or assisted living facilities were included. In total, 513 interviews were included in the analysis. Participation rate was 34%. The full study protocol is available elsewhere [24].

The authors assert that all procedures contributing to this work comply with the ethical standards of the relevant national and institutional committees on human experimentation and with the Helsinki Declaration of 1975, as revised in 2008. The study received ethical approval from the ethical committee of Ghent University/University Hospital (B670201837542) and was conducted according to the WHO ethical and safety recommendations for SV research [25]. As recommended by these guidelines, the study was presented as the “Belgian study on health, sexuality and well-being.” Written informed consent was obtained from all participants. After participation, participants received a brochure with the contact details of several helplines. 

### 2.2. Definitions and Measures

This study was a part of a national SV prevalence study in the Belgian population between 16 and 100 years old [23]. The questionnaire consisted of several modules, including sociodemographic characteristics, sexual health and relations, mental health, and sexual victimisation. The questions in these modules were identical across all age groups. 

Mental health was measured using international scales, which were validated in several age groups. Depression was measured using the Patient Health Questionnaire (PHQ)-9 (Cronbach’s alpha (α) = 0.737) [26] and anxiety using the General Anxiety Disorder (GAD)-7 (α = 0.827) [27]. Both scales assessed symptoms in the two weeks before the interview. PTSD was assessed with the Primary Care PTSD Screen for DSM-5 (PC-PTSD-5) (α = 0.572) [28], which questioned symptoms in the month before the interview. Resilience was estimated using the Brief Resilience Scale (BRS) (α = 0.821) [29]. Additionally, participants were asked about self-harm and suicide attempts both during their lifetime and in the past 12 months. Social support was measured by number of confidants. 

SV was defined according to the WHO definition, which encompasses different forms of sexual harassment without physical contact, sexual abuse with physical contact but without penetration, and (attempted) rape [1]. As a result of recent insights in the field of SV in older adults, this definition was expanded to include sexual neglect [3], which was measured as “touching in care” due to the absence of a standardised measure (See Appendix A). We used behaviourally specific questions (BSQ) to assess lifetime and past 12-month SV experiences. The SV items were based on existing surveys [30,31,32] and adapted to the Belgian social and legal context [33]. To calculate lifetime and past 12-month sexual victimisation, we created dichotomous variables out of all 17 SV items. Due to a high level of multicollinearity between hands-off and hands-on SV (Variance Inflation Factor (VIF) > 4) [34], they were not added separately in the analyses but were combined into the variables lifetime and past 12-month SV. Lifetime SV is defined as exposure to any of the 17 SV items (Yes/No) provided in Appendix A during the lifetime of the participant (age 0 till time of the survey).

### 2.3. Statistical Analysis 

Statistical analysis was performed using SPSS Statistics 26 [35] and R version 3.6.3 [36]. Descriptive statistics (means, standard deviations, and percentages) were calculated for all variables. Outcome variables were described in a categorical way, as this is how the scales are used in clinical practice. We conducted a bivariate analysis to compare the proportion of sexual victimisation within the different categories of depression, anxiety, and PTSD using a chi-square test or a Fisher’s exact test if the assumptions of the chi-square test were not met. 

To assess the association between lifetime sexual victimisation and depression, anxiety, and PTSD, we performed stepwise linear regression analyses for each outcome variable separately. In each model, we used the score on the PHQ-9, the GAD-7, and the PC-PTSD-5 scale as a continuous outcome variable. The assumption of normally distributed residuals was checked through a Q-Q plot and a histogram (see Appendix B). Moreover, the kurtosis (<5) and skewness (<2) were checked for all outcome variables and were considered acceptable [37,38]. In each analysis, main effects were added and/or deleted stepwise based on selection criteria (*p* < 0.05). The multi-collinearity assumption of multivariate regression analyses was tested for the main effects with the VIF and indicated no violation. Given the low numbers for suicide attempts and self-harm in the past 12 months, we did not perform a regression analysis on these variables. We constructed cross-product terms to further explore the moderating effects of past 12-month SV, sex at birth, age, sexual orientation, resilience, education level, perceived health status, care dependency, and social support in the relationship between lifetime sexual victimisation and depression, anxiety, or PTSD. The potential moderators were dichotomised. Continuous variables were dichotomised based on the mean value. Afterwards, cross-product terms were created between each of the binary moderators and lifetime SV. Again, stepwise linear regression analyses were conducted. In each analysis, the cross-product terms were added and/or deleted stepwise together with their main effects based on selection criteria (*p* < 0.10). Interactions were further explored if *p* < 0.10 to account for possible underpowering of the sample. In that case, separate regression analyses were run to further analyse the interaction effect (e.g., separate analyses for higher versus no/lower education). In the final model, only significant (*p* < 0.05) main effects and interaction effects were retained. *p*-values, coefficients, and 95% confidence intervals (CI) are presented.

## 3. Results

### 3.1. Descriptive Statistics

The study sample consisted of a representative sample of the Belgian population aged 70 years and older (*n* = 513) [24]. The mean age was 79 years (SD: 6.4 yrs, range 70–99 yrs), 58.3% was female, 7.4% identified themselves as non-heterosexual, 89.8% was community-dwelling, 31.2% had completed higher education, 26.1% labelled their financial situation as difficult, and 48.3% reported low social support. Regarding health status, 45.8% indicated that they suffered from an illness or disability limiting their daily activities, 46.4% was care-dependent, and 26.5% reported low resilience. Over 44% of the participants experienced SV during their lifetime and 8.4% in the past 12 months. A detailed description of the different forms of sexual victimisation has been published elsewhere [4]. More information on the characteristics of the study sample can be found in Table 1.

Regarding mental health status (see Table 2), 19.9% reported mild, 5.1% moderate, and 2.5% severe depressive symptoms; and 17.5% mild, 4.5% moderate, and 3.5% severe anxiety symptoms during the two weeks prior to the interview. During the month prior to the interview, 5.7% suffered from symptoms of PTSD. We found a significant difference in level of depression, anxiety, and PTSD between SV victims and non-victims (*p* < 0.05). Over 5% attempted suicide during their lifetime and 1.6% in the past 12 months; 2.1% reported self-harm during their lifetime and 1.4% in the past 12 months. For both suicide attempts and self-harm, we found no significant difference between SV victims and non-victims.

### 3.2. Association between Lifetime SV and Depression, Anxiety, and PTSD

The linear regression analysis (Table 3) showed that having experienced lifetime SV, financial difficulties, having a chronic illness or disability limiting daily activities, being care dependent, and having fewer than three confidants were associated with depression in old age. Exposure to SV in the past 12 months, sex at birth, age, sexual orientation, education level, and resilience were not associated with depression in our sample. Anxiety was associated with lifetime exposure to SV, a higher education, financial difficulties, a chronic illness or disability limiting daily activities, being care dependent, fewer than three confidants, and low resilience. Exposure to SV in the past 12 months, sex at birth, age, and sexual orientation were not associated with anxiety in our sample. Financial difficulties were associated with PTSD. Lifetime SV was only associated with PTSD in respondents with a chronic illness/disability or no/lower education. Exposure to SV in the past 12 months, sex at birth, age, sexual orientation, care dependency, social support, and resilience were not associated with PTSD in our sample. 

The direction of the association between lifetime SV, depression, and anxiety are shown in the first two graphs of Figure 1. The last two graphs show the interaction effect between lifetime SV and education level and between lifetime SV and perceived health status in the case of PTSD.

## 4. Discussion

In this paper, we present the results of a study on the mental health impact of sexual victimisation in a representative sample of 513 older adults living in Belgium. We established the prevalence of depression, anxiety, PTSD, suicide attempts, and self-harm in the Belgian population of 70 years and older. Moreover, we studied the association between lifetime sexual victimisation and depression, anxiety, PTSD, suicide attempts, and self-harm in later life. Furthermore, we examined the moderating effects of sexual victimisation in the past 12 months, sociodemographic characteristics, health status, social support, and resilience in the relationship between lifetime sexual victimisation and depression, anxiety, and PTSD. 

### 4.1. Mental Health Problems Are Common in Older Adults in Belgium

Our results confirm that mental health problems are common in older adults [5]. Almost 20% of participants reported mild depressive symptoms, 5.1% reported moderate depressive symptoms, and 2.5% suffered from severe depressive symptoms. Over 17% reported mild, 4.5% moderate, and 3.5% severe anxiety symptoms, and 5.7% suffered from PTSD. Previous studies found similar numbers regarding depression and anxiety [6,7] but lower prevalence numbers for PTSD compared to our study [8]. However, our PTSD prevalence numbers even seem underestimated, as some studies indicate subclinical ranges of PTSD in older adults up to 17% [39]. Several studies have shown that different symptoms clusters may exist between younger and older adults with PTSD. Older adults may report more somatic and affective complaints [39,40], which are not questioned in the PC-PTSD-5, the PTSD measurement instrument used in our study. Moreover, the use of different definitions and measurement instruments makes comparing PTSD prevalence rates between studies difficult. While previous studies defined PTSD according to the criteria of the Diagnostic and Statistical Manual of Mental Disorders IV (DSM-IV), we used the DSM-5 PTSD criteria, which could influence prevalence rates. Although studies in younger populations showed no difference in prevalence rates when using the DSM-IV versus the DSM-5 criteria [41,42], this has not yet been researched in older adults. Additionally, studies using the DSM-5 criteria use different measurement instruments, such as the Clinician-Administered PTSD scale for DSM-5 (CAPS-5) [43], the PTSD Checklist for DSM-5 (PCL-5) [44], or the PC-PTSD-5 [28]. This complicates the comparison of PTSD prevalence rates between different studies using the same DSM-5 definition. Therefore, more research comparing different measurement instruments and their validity in older adults is needed. Regarding suicide attempts and self-harm, we found that 1.6% of our study population reported a suicide attempt and 1.4% self-harm in the past 12 months. Similar rates were found by the Flemish Centre for Suicide Prevention (VLESP) in the same period [45]. Although our results and previous research shows low numbers for suicide attempts and self-harm, it is known that older adults complete suicide proportionally more than any other age group [9,45]. However, information on SV and mental health of older adults who completed suicide were not available in our study. 

In addition to the high prevalence of mental health disorders, our study confirms previous findings indicating that financial difficulties, suffering from a chronic illness or disability, being care dependent, and having low social support are associated with adverse mental health outcomes in later life [7,9,46]. Therefore, we endorse previous calls to invest in psychosocial interventions that combat isolation in older adults, including befriending programmes and peer support schemes [47]. Although previous studies link a low education level to adverse mental health outcomes in later life [7,9,46], our study shows mixed results. In our sample, older adults with a low education level showed fewer anxiety symptoms compared to highly educated older adults. For depression and PTSD, we found no link between education level and symptom severity. Concerning resilience, our results show that resilient older adults report fewer anxiety symptoms. Whereas previous studies indicated a link between resilience and depressive symptoms in older adults [47], we found no association in our sample. Since the definition and measurement of resilience continues to be the subject of debate [48], and new scales for resilience in older adults were developed and tested very recently [49], the different results could possibly be explained by different measurement instruments. More research comparing the reliability of different measurement instruments for resilience in older adults is needed to draw conclusions concerning its moderating effect on mental health outcomes in later life. 

### 4.2. Lifetime Sexual Victimisation Is Linked with Depression, Anxiety, and PTSD in Older Adults 

The second objective of this paper was to check whether lifetime sexual victimisation was associated with depression, anxiety, PTSD, suicide attempts, and self-harm in late life. Both the bivariate and the regression analysis showed an association between lifetime sexual victimisation and current depression and anxiety in old age. For PTSD, we found an association in respondents with a chronic illness/disability or no/lower education. Lifetime sexual victimisation and suicide attempts or self-harm in the past 12 months were not significantly associated. This could be due to the low prevalence rates for suicide-attempts and self-harm limiting the power of our analysis. As discussed earlier, studies show low numbers of suicide attempts in older adults [45]. Since sexual victimisation is a known risk factor for suicide attempts, and older adults are more likely to die after a suicide attempt [9,45], we can partly assume that we could not establish a link between sexual victimisation and suicide attempts in our sample since possible victims might have died by suicide. 

### 4.3. Past 12-Month Sexual Violence, Social Support, and Resilience Do Not Moderate the Relationship between Lifetime Sexual Victimisation and Depression, Anxiety, and PTSD

Thirdly, this paper examined the moderating effect of past 12-month SV, sociodemographic characteristics, health status, social support, and resilience in the relationship between lifetime sexual victimisation and depression, anxiety, and PTSD in older adults. In our sample, past 12-month SV did not moderate the relation between lifetime sexual victimisation and any of the mental health outcomes, suggesting that exposure to SV earlier in life had such an important impact on the victims’ mental health that the impact of a recent SV event made no difference. Regarding sociodemographic characteristics, we found a moderating effect of education level on the relationship between lifetime sexual victimisation and PTSD. Similarly, health status moderates the relationship between lifetime sexual victimisation and PTSD. Although previous research found that social support in the acute phase after exposure to SV is crucial to decrease adverse mental health outcomes [15,16], our study indicated that social support does not influence long-term mental health outcomes after SV. We assume this could be explained by the limited help-seeking behaviour of older adults upon sexual victimisation. Recent research has shown that 60% of older SV victims never disclosed their experiences to their informal network, and 94% never sought professional help [50]. Similar to younger victims [51], older SV victims experience many barriers to disclose SV experiences or seek professional help [20,50]. We hypothesize that social support is only beneficial for long-term mental health outcomes following sexual victimisation when the confidants are aware of the victim’s SV history. More research on the interaction between social support, SV disclosure, and mental health outcomes in older adults is needed to confirm this hypothesis. Furthermore, resilience did not moderate the effect between lifetime sexual victimisation and depression, anxiety, and PTSD in our sample. Previous studies on the effect of resilience showed mixed results. Some found that resilient subjects may be partly protected against the long-term mental health effect of SV [17], but others could not establish high resilience as a protective factor against adverse mental health outcomes after SV [18]. More research on the association between sexual victimisation and resilience on mental health outcomes in older adults is warranted to clarify this. 

### 4.4. Limitations

Our study has several limitations. First, our study applied a cross-sectional design, which makes it impossible to establish a causal relation between sexual victimisation and mental health outcomes in later life. Although a longitudinal study described an association between childhood sexual abuse and internalising disorders in old age [12], another study has shown that people with severe mental illness (SMI) experience substantially more SV during their lifetime compared to those who do not suffer from SMI [52]. Second, sexual victimisation was reported retrospectively by the participants. Although there is evidence that the magnitude of associations with mental disorders does not differ according to whether sexual victimisation was reported prospectively or retrospectively [53], evidence from a prospective study would further strengthen our conclusions. Third, we applied a broad definition of SV, including both hands-off and hands-on SV. One might assume that different forms of SV might have a different mental health impact. However, due to a high level of multicollinearity between hands-off and hands-on SV (VIF > 4) [34], our study was not able to differentiate between the mental health impact of lifetime hands-off versus hands-on SV. Fourth, although several studies have indicated that the age of first exposure to different types of violence, including SV, may be associated to differential risks of various mental health outcomes [54,55,56], our study does not distinguish between lifetime sexual victimisation during childhood and adulthood. We recommend future studies to add a follow-up question in the SV module regarding the age of first SV exposure. This would allow us to differentiate the mental health impact of SV exposure during childhood and adulthood. Finally, seeking (in)formal help upon sexual victimisation was not included as a confounding variable in our model. Although seeking (in)formal help might have a positive effect on long-term mental health outcomes following SV, research has shown that very few older SV victims had disclosed their experiences or sought professional help [50]. Therefore, we assume a limited impact of help-seeking upon SV on mental health outcomes in old age. 

### 4.5. Recommendations for Future Research and Clinical Practice

Despite its limitations, this study improves the current understanding of the mental health impact of SV in older adults. Being the first to apply a life course perspective in the analysis, including sexual victimisation during childhood, adulthood and old age, and using a broad definition of SV, this study offers a unique understanding of the mental health impact of SV in late life. To strengthen our conclusions, studies using the same approach and SV definition in older adults from different countries and different cultural backgrounds are warranted. Moreover, in order to provide tailored care for older victims of SV, mental health care should be better aligned with the needs of older victims of SV. Therefore, incorporating a routine inquiry about sexual victimisation, as proposed by the WPA [21], should become the gold standard of care for older adults who present with mental health problems. However, since health care workers rarely feel comfortable discussing SV with older adults [22], capacity building for healthcare workers and the development of clinical guidelines and care procedures seem of the utmost importance.

## 5. Conclusions

Lifetime sexual victimisation is linked to depression and anxiety in older adults in Belgium and is associated with PTSD in older adults with a chronic illness/disability or no/lower education. Our findings confirm the long-lasting mental health impact of sexual victimisation and show the need for tailored mental health care for older victims of SV. Therefore, more research on the specific needs of older SV victims regarding mental health care is needed. In addition, professionals working with older adults urgently need capacity building regarding SV and its mental health impact. Furthermore, the development of clinical guidelines and care procedures seems particularly important.

## Figures and Tables

**Figure 1 ijerph-19-02803-f001:**
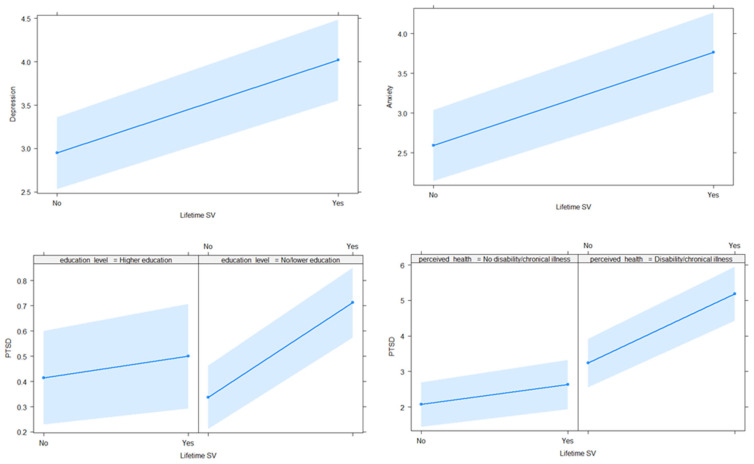
Association between lifetime sexual victimisation and depression, anxiety, and PTSD.

**Table 1 ijerph-19-02803-t001:** Descriptive statistics of sociodemographic characteristics, health status, social support, resilience, and sexual victimisation (*n* = 513).

Variable		*n* (%)
Sex at birth	Female	299 (58.3)
	Male	214 (41.7)
Age ^a^	70–79 yrs	283 (55.2)
(mean 79 yrs)	80–89 yrs	201 (39.2)
	90–99 yrs	29 (5.7)
Sexual orientation	Heterosexual	475 (92.6)
	Non-heterosexual ^b^	38 (7.4)
Living situation	Community-dwelling	461 (89.8)
	Assisted-living facility	25 (4.9)
	Nursing home	27 (5.3)
Education level	No/lower education ^c^	353 (68.8)
	Higher education	160 (31.2)
Financial status ^d^	Easy	377 (73.5)
	Difficult	134 (26.1)
Social support ^e^	Low	246 (48.3)
	High	265 (51.7)
Perceived health status	No disability/chronical illness	278 (54.2)
	Disability/chronical illness	235 (45.8)
Care dependency	Yes	238 (46.4)
	No	275 (53.6)
Resilience ^f^	Low	136 (26.5)
	Normal	319 (62.2)
	High	58 (11.3)
Sexual victimisation	Lifetime	227 (44.2)
	Past 12 months	43 (8.4)

^a^ Calculated based on the participant’s year of birth. ^b^ This group contains participants who labelled themselves as homosexual, bisexual, pansexual, asexual, or other. In this last group, several participants labelled themselves as “normal.” Since it was not clear whether they had difficulties understanding the different terms defining sexual orientation or whether they indeed identified their sexual orientation as “other,” we decided to classify these participants as non-heterosexual based on their answer. ^c^ This group contains participants who had no formal education or completed primary education, secondary education, technical or vocational education, or religious school. ^d^ Measured through the ability to make ends meet. The four answer categories, “with great difficulty”, “with some difficulty”, “fairly easily”, and “easily,” were dichotomised into “difficult” and “easy.” ^e^ Measured by number of confidants. The average number of confidants was 3. Fewer than 3 confidants corresponds to low social support, while 3 or more confidants corresponds to high social support. ^f^ Measured using the Brief Resilience Scale (BRS): low (1.00–2.99); normal (3.00–4.30); high (4.31–5.00).

**Table 2 ijerph-19-02803-t002:** Descriptive statistics of depression, anxiety, PTSD, suicide attempts, self-harm, and sexual victimisation.

Item	Scale	Outcome	*N* Total	% Total	% SV	Chi-Square Test
Depression	PHQ-9	No	371	72.5	41.2	*p* = 0.047
(*n* = 512)	(α = 0.737)	Mild	102	19.9	47.1	
		Moderate	26	5.1	65.4	
		Moderately severe/Severe	13	2.5	61.5	
Anxiety	GAD-7	No	382	74.5	40.8	*p* = 0.043
(*n* = 513)	(α = 0.827)	Mild	90	17.5	51.1	
		Moderate	23	4.5	60.9	
		Severe	18	3.5	61.1	
PTSD	PC-PTSD-5	Yes	29	5.7	65.5	*p* = 0.018
(*n* = 512)	(α = 0.572)					
Suicide attempts	NA	Lifetime	27	5.3	51.9	*p* = 0.414
(*n* = 513)		Past 12 months	8	1.6	37.5	*p* = 1.00
Self-harm	NA	Lifetime	11	2.1	36.4	*p* = 0.762
(*n* = 512)		Past 12 months	7	1.4	28.6	*p* = 0.471

α, Cronbach’s alpha; SV, sexual victimization; PTSD, posttraumatic stress disorder. % SV: proportion of sexual victimisation within the different categories of depression, anxiety, PTSD, suicide attempts, and self-harm. Patient Health Questionnaire-9 (PHQ-9): mild (5–9), moderate (10–14), moderately severe–severe (≥15); General Anxiety Disorder-7 (GAD-7): mild (5–9), moderate (10–14), severe (≥15); Primary Care PTSD Screen for DSM-5 (PC-PTSD-5): yes (≥3).

**Table 3 ijerph-19-02803-t003:** Association of lifetime sexual victimization with depression, anxiety, and PTSD.

	Depression (PHQ-9)			Anxiety (GAD-7)			PTSD (PC-PTSD-5)		
	Coeff.	95% CI	*p*-Value	Coeff.	95% CI	*p*-Value	Coeff.	95% CI	*p*-Value
**Main effects**									
Lifetime SV	1.074	0.451, 1.697	0.001	1.173	0.504, 1.841	0.001	0.284	0.129, 0.439	<0.001
Education level	--	--	--	−0.880	−1.615, −145	0.019	0.039	−0.129, 0.208	0.647
Financial status	1.248	0.544, 1.953	0.001	1.287	0.518, 2.056	0.001	0.211	0.034, 0.388	0.019
Perceived health status	1.988	1.324, 2.651	<0.001	1.345	0.633, 2.057	<0.001	0.225	0.069, 0.380	0.005
Care dependency	1.210	0.548, 1.872	<0.001	1.259	0.544, 1.974	0.001	--	--	--
Social support	0.823	0.201, 1.445	0.010	0.990	0.320, 1.659	0.004	--	--	--
Resilience	--	--	--	0.751	0.075, 1.426	0.029	--	--	--
**Cross-product terms**									
Lifetime SV × education level	--	--	--	--	--	--	0.318	−0.017, 0.652	0.062
Lifetime SV × perceived health status	--	--	--	--	--	--	0.310	0.000, 0.620	0.050
**Interaction effects**									
Lifetime SV by education level	--	--	--	--	--	--			
Association in higher education	--	--	--	--	--	--	0.101	−0.178, 0.380	0.477
Association in no/lower education	--	--	--	--	--	--	0.356	0.165, 0.547	<0.001
Lifetime SV by perceived health status	--	--	--	--	--	--			
Association in no chronic illness/disability	--	--	--	--	--	--	0.154	−0.026, 0.334	0.093
Association in chronic illness/disability	--	--	--	--	--	--	0.422	0.155, 0.689	0.002

SV, sexual victimization; PTSD, posttraumatic stress disorder; PHQ-9, Patient Health Questionnaire-9; GAD-7, General Anxiety Disorder-7; PC-PTSD-5, Primary Care PTSD Screen for DSM-5; Coeff, coefficient; CI, confidence interval. Note. In this table, we describe the final linear regression models for depression, anxiety, and PTSD, in which only significant (*p* < 0.005) main and interaction effects were retained. Other possible confounders included in the model were sex at birth, age, and sexual orientation. The coefficient describes the variability of the outcome variable in comparison to a reference value. For lifetime and past 12-month SV, this is “no SV”; for sex at birth, this is “male”; for age, this is “< 79 years”; for education level, this is “higher education”; for financial status, this is “easy”; for perceived health status, this is “no chronic illness/disability”; for care dependency, this is “no care dependency”; for social support, this is “high ≥ 3”; for sexual orientation, this is “heterosexual”; and for resilience, this is “high ≥ 3.40 (mean)”.

## Data Availability

By request from the corresponding author.

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
