# Peer review of "“Time Does Not Heal All Wounds”: Sexual Victimisation Is Associated with Depression, Anxiety, and PTSD in Old Age"

_ijerph, 2022, doi:10.3390/ijerph19052803_

Round 1

Reviewer 1 Report

Dear Authors; this is a novel study on the association of lifetime SV and three key mental health outcomes. It needs some improvements to arrive to journal standard listed below. Regards,

P.S.

[1] Statistical:

1-1 Lifetime SV: this variable is defined almost unclear. As medical statistician, it took me ten minutes back and forth to figure it out !   Please add the following definition in the manuscript Methods Section foir the clarification of the readers:

Definition. Life time Sexual Violence is defined as incidence status(Yes/No) of any item of the set of seventeen items provided in the Appendix at lifetime of the participant (age 0 till time of the survey). 

1-2 "Seeking therapy" and "age" are potential confounders for this study. Justify lack of their use in the multivariate linear regression or add it to the limitations of the study.

1-3 The multivariate linear regression had "Depression", "Anxiety" and "PTSD" as its outcomes. Please add report how did you check normality of these three key outcomes ?   How did you check the association is "linear" and not "non-linear"?   It is good you add your boxcox test result to the Methods section.

1-4 Please add two citations (one for each) for the statistical softwares packages used in the study in the references section: "SPSS", "R" in line 138.

[2] Writing

2-1 Please check the entire manuscript grammar and punctuation: line 129:  "Appendix 1"  ---> "Appendix A";  line 352:    add punctuation ". Another limitation ...". 

2-2 Please add a list of abbreviation used in the manuscript for your readers referral right before References section.  Order it alphabetically. Example:

PTSD: Posttraumatic Stress Disorder

Author Response

The authors would like to thank the editorial board of the International Journal for Environmental Research and Public Health for their interest in our manuscript. We also would like to thank the reviewers and the editor for the useful remarks and suggestions. In the document attached we enumerate how we handled each of the suggestions of the reviewers in the revision of our manuscript. The actual revisions are marked by means of track changes in the text. We sincerely hope that the adaptions are sufficiently forthcoming to the journal’s demands for publication of this revised manuscript. We remain at your entire disposal for any further indications or clarifications.

Reviewer 2 Report

The topic of this article is highly relevant for the field. The authors use a lifespan perspective in a sample of older adults to examine 1) the prevalence of mental health disorders (depression, anxiety, PTSD), suicide attempts and self-harm, as well as its moderators 2) connections between lifetime sexual victimization, mental health disorders (depression, anxiety, PTSD), suicide attempts and self-harm, as well as its moderators, and 3) the moderating effect of these moderators on  associations between lifetime sexual victimization and mental health disorders. A cross sectional sample of 513 older adults from Belgium completed surveys assessing these variables. Findings revealed that lifetime exposure to SV was linked to mental health disorders, but not to past-year suicide attempts or self-harm. 

Overall, the manuscript is scientifically sound and the methodology appears appropriate to investigate the research questions. The methods are described in enough detail that the manuscript’s results can be reproduced. The materials in the Appendix are especially helpful.

The figures, tables, and materials contained in the Appendix seem appropriate. They display the data properly and appear easy interpret and understand. The data appear to be interpreted appropriately and consistently throughout the manuscript, with some limitations acknowledged.

However, other important limitations are not acknowledged, and this is my primary area of concern/critique. The authors are transparent about their approach, which used an extremely broad definition of SV. There should at least be a detailed acknowledgement of how different types of SV exposures may be associated with different mental health outcomes, and how these relationships could have impacted the results. Lifetime SV exposure fails to differentiate between SV first experienced during childhood versus as an adult. SV experienced early in life can impact the developing brain, mind, and body in different ways  For example, there is research on how the physiological fight/flight vs. freeze responses can be activated differently in response to different types of trauma exposures, with factors like, proximity of threat and physical immobilization playing a substantial role. It would follow that an individual may have different physiological reactions to undesired sexual innuendo versus forced penetrative and violent rape. Fight/flight responses are associated with hyperarousal and the more traditional PTSD symptoms, while the freeze response is associated with hypoarousal and symptoms marked by dissociation, and these can lead to different presentations of PTSD and other mental health issues across the lifespan. The DSM-5 includes a dissociative subtype – however, dissociative symptoms are not explicitly measured as part of the Primary Care PTSD Screen for DSM-5. This is particularly relevant for this article, given that dissociative symptoms can uniquely contribute to suicidal thoughts and behaviors, as well as self-harm, as detailed in the concept of the Acquired Capability for Suicide, part of the Interpersonal Theory of Suicidal Behavior. In addition, there is research from the United States indicating that over 90% of children involved in the Illinois child welfare system do not qualify for a PTSD diagnosis. The narrow scope of the PTSD diagnosis in the DSM-5 has been rightly critiqued, particularly for its failure to account for the developmental aspects of early life trauma, such as SV, as well as for it’s lack of attention to a more complex symptom presentation, the latter of which were previously recognized in the DSM-IV’s Complex-PTSD sub-category. These issues need to be acknowledged and described in the limitations section, and a justification for using a broad definition of SV should be provided. Also, the authors mention in the Discussion section (lines 271-275) that previous studies have used the DSM-IV criteria to define PTSD. It would be helpful for them to list the name of the measure(s) used, in order for readers to understand the scope and type of symptoms measured.

The majority of references are from the last 12 years. If studies from the last 5 years are available to replace these, they should be incorporated, with the understanding that the study of sexual violence among older adults is a very specific area of research, and as such, there may be a lack of relevant recent studies.

Self-citations are not excessive, and they appear to be appropriate.

The ethics and data availability statements appear to be adequate.

Line 207 – change to Cronbach’s “alpha”

Author Response

(The authors gave the same response as above.)
